# Impact of alternative Non-Pharmaceutical Interventions strategies for controlling COVID-19 outbreak in Bangladesh: A modeling study

**Shafiun Nahin Shimul**[1,2]*, **Mofakhar Hussain**[3], **Abul Jamil Faisel**[4], **Syed Abdul Hamid**[1], **Nasrin Sultana**[1], **Md Abdul Kuddus**[5]

**1** Institute of Health Economics, University of Dhaka, Dhaka, Bangladesh, **2** CDC Foundation Postdoctoral Fellow, Georgia State University, Atlanta, Georgia, **3** Institute of Health Policy, Management and Evaluation University of Toronto, Toronto, ON, Canada, **4** President-Elect, The Public Health Association of Bangladesh & Institute of Health Economics, University of Dhaka, Dhaka, Bangladesh, **5** Department of Mathematics, University of Rajshahi, Rajshahi, Bangladesh

* shafiun.ihe@du.ac.bd

**Data Availability Statement:** The data are available on Figshare (DOI: 10.6084/m9.figshare.20010209. v1).

## Abstract

The COVID-19 pandemic has been a major health concern in Bangladesh until very recently. Although the Bangladesh government has employed various infection control strategies, more targeted Non-Pharmaceutical interventions (NPIs), including school closure, mask-wearing, hand washing, and social distancing have gained special attention. Despite significant long-term adverse effects of school closures, authorities have opted to keep schools closed to curb the spread of COVID-19 infection. However, there is limited knowledge about the impact of reopening schools alongside other NPI measures on the course of the epidemic. In this study, we implemented a mathematical modeling framework developed by the CoMo Consortium to explore the impact of NPIs on the dynamics of the COVID-19 outbreak and deaths for Bangladesh. For robustness, the results of prediction models are then validated through model calibration with incidence and mortality data and using external sources. Hypothetical projections are made under alternative NPIs where we compare the impact of current NPIs with school closures versus enhanced NPIs with school openings. Results suggest that enhanced NPIs with schools opened may have lower COVID-19 related prevalence and deaths. This finding indicates that enhanced NPIs and school openings may mitigate the long-term negative impacts of COVID-19 in low- and middle-income countries. Potential shortcomings and ways to improve the research are also discussed.

## Introduction

It has been years since the start of the COVID-19 pandemic in Bangladesh (as of June 2021). The trajectory of the pandemic shows cases and deaths have surged, decreased, and have surged again. In the middle of 2021, concerns about the invasion of the COVID-19 virus of concerns (Indian variant, B16117) led the government to put in place restrictions and lockdown and then the reported cases were about 800,000, and reported deaths were about 12,800.

**Funding:** The author(s) received no specific funding for this work.

**Competing interests:** The authors have declared that no competing interests exist.

In the wake of an infectious disease epidemic, governments implement Non-Pharmaceutical Interventions (NPI) to reduce the spread of the disease with the objective to lower rate of infections and deaths. Typical NPI involves improvement in hygiene (hand washing, mask-wearing) and reduced inter-personal contacts (social distancing, school closures, travel limits, stay-at-home orders, isolation and quarantine). Since the start of COVID-19 in Bangladesh, all of these measures have been attempted with varying degrees of success.

School closure has been the longest and strictest among all the NPI measures implemented in Bangladesh, with all schools being closed for almost 18 months since March, 2020. School closure reduces inter-personal contact, especially among similar aged cohorts; it also increases contacts at home by 20%. The net effect is a lower level of contacts and lower levels of transmission of infection.

Although on-line learning opportunities have been announced, surveys found that only 40% have access to such opportunities [1]. For more than 12 months, 3.7 million students, from pre-kindergarten to university, have been deprived of learning and educational opportunities. This is expected to have several negative consequences. Schools provide development of vital intellectual, social and physical skills critical for survival and growth. A year without such skills can have long-term negative impacts on learning. A significant percentage of students may never return to school after such long absence. In another circumstance, a large number of underage teenage girls may be married off, triggering a sequence of negative health effects.

School opening may result in increased transmission cycle but at the same time may result in more teachers and students becoming aware of the importance of infection control measures such as mask-wearing and handwashing. Spreading this message and practicing at the community level, where schools are, at a consistent basis may result in sustained use of masks and hand washing. Thus, over time, infection transmission may slow down—though it may sound ironical—due to school opening. However, there is a strong opposition to school opening, arguing that the benefit of school closure outweighs the cost. This reasoning is based on the observations that school closures result in fewer deaths compared to scenarios when schools are open. However, if other controls measures are not in place, even with school closures, the rate of death may not witness a declining trend.

Published studies have reviewed the impact of school closures on COVID-19 related transmission of infection and deaths. Most of these studies have been done in developed countries. These studies reveal limited [2, 3] or no [4] to mixed [5] effects. Other studies reveal that school closures and other NPI may also have contributed to lower COVID-19 cases and deaths [6]. Further, the previous research on 2003 SARS virus in China, Hong Kong, and Singapore reveals that school closures have had a limited impact on transmissions of infection [5].

However, extended school closures have long-term negative impacts on children in multiple ways, including learning loss, school dropout, reduced skills development, child marriages, and increased mental stress. These negative effects are especially high in resource poor settings where alternatives to in-person schooling are limited resulting significant "learning gap" for children in low-income families [7, 8].

A study by World Bank (WB) estimates that school closures in Bangladesh could result in a loss of 0.6–0.9 years of learning adjusted schooling for an average child which in turn could contribute to long timer economic loss in terms of lost earning [9]. WB also estimates 3–13% of annual GDP could be lost due to school closures [10]. In addition, child marriage is expected to increase as well.

A recent analysis of effects on school closures in Bangladesh reveal that only about 10% of school going children had access to distance learning, 19–25% experienced learning loss (about 6 million students), 12% experienced mental stress. In a technical note published by WHO and UNICEF on September 2020, noted that extra school closure may provide minimal

health benefit to children due to COVID-19 yet result in significant negative impacts on health and economy [11].

Given the mixed or limited effects of school closures on COVID-19 transmission and deaths and clear academic, health and economic negative effects, re-opening schools have become urgent. Yet researchers also argue that COVID-19 may be a long-term phenomenon with uncertain recurrences. The strength and versatility of the COVID-19 virus since the start of the pandemic indicates that the duration of the pandemic may extend beyond 2022 [12]. At the same time, in Bangladesh, the introduction of variants of concern and lack of sustained vaccination programs indicate that the risk of the pandemic will remain beyond 2021. Thus, efforts to open schools even while the pandemic is on-going may need to be considered. In this study, we evaluate whether school re-opening along with higher levels of NPI will be more effective than lower levels of NPI along with school closures on COVID-19 related infections and deaths. We employ a mathematical COVID-19 model to evaluate the impact. All necessary data, including incidence, prevalence, mortality, total population, the number of hospital beds and ICU beds, contact rate, parameters related to hospitalization, and time-varying levels of NPI parameters, are analyzed to meet our objectives.

This paper is constructed as follows: the following section describes the method and data. After that, we present the results of this study. A brief discussion, recommendation, and concluding remarks finalize the paper.

## Method

In this study, we use a disease modeling technique to estimate the impact of different hypothetical interventions on COVID-19 infections and deaths. For this purpose, we use a compartemntal model developed by CoMo Consortium (Univesity of Oxford), with individuals classified according to current disease status (see Fig 1) and transmission between homogeneous mixing population [13]. It is worth noting that authors of this manuscripts have used the model developed by CoMo Consortium. Even though authors were part of CoMo consortium to implement this model for Bangladesh context, the authors of this manucripts were not directly involved in developing the core or base model.

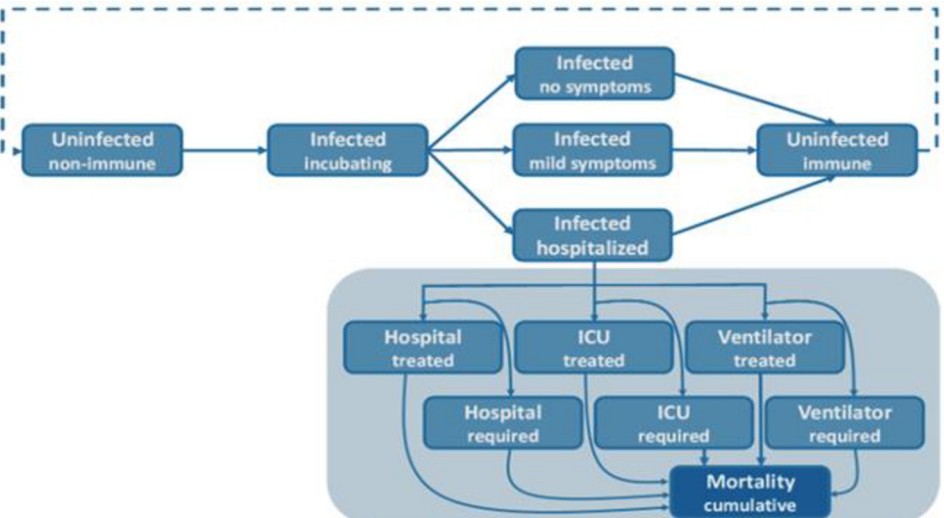

**Fig 1. State transition in the model: Individuals in the compartmental model are classified into Uninfected non-immune, infected incubating, infected no symtoms, infected mild symptoms, infected hospitalized and uninfected immune.**

The CoMo model is a deterministic, age-structured SEIR model. It incorporates different treatment categories (hospital, ICU and ventilator). The Fig 1 provides a schema of the CoMo model. Briefly, all individuals in the model start as susceptible to COVID-19 infection, and enter the incubating infected state upon an effective contact with an infectious person. After a latent period, incubating infected individuals become infectious, either without symptoms, mild symptoms infection or with a hospitalized infection. After the infectious period, individuals enter the uninfected state due to recovery or isolation and some of the uninfected individuals move to the susceptible state due to the rate of loss of immunity. However, some of the hospitalized infected individuals move to the hospital treated, ICU treated and ventilator treated compartment and they require hospital, ICU and ventilator respectively. All of the hospital treated, ICU treated and ventilator treated may move to the mortality state. The model flow diagram is presented in Fig 1. A more detailed version is described in [13].

The model developed by the CoMo Consortium was deployed into a web-based application (available here https://comomodel.net/) which end users can run using country specific measures.

The projection exercise begins with uploading country specific data in a template and uploading into the CoMo hosted website. Once uploaded into the website the model can be run and intervention and other parameters can be adjusted to align historical actuals with the projections for the time period for which actual data is available. In this calibration stage, the model parameters are changed to ensure the actual cases and deaths are closely aligned with those projected by the model for the same time period. The process also allow user to input hypothetical interventions in the future, use of which results in two different scenarios in the future: baseline and hypothetical. After the model is run, the output includes projected cases (reported and unreported), levels of hospitalizations and deaths and provides summative parameters such as prevalence of COVID-19, time specific rate of transmission (R0). An excel file containing daily measures into the last day of the projection period is also produced and can be downloaded.

For the purpose of this study, we use CoMo Model Version 17. It is worth nothing that all the codes, models, apps are publicly available; hence, replicability of the results is warranted. The supplementary (S1: Replication Process and Parmeter Values in S1 File) provides input data sets and steps used in the web-based CoMo Consortium model to create the results reported in this study. Following those steps anyone can reproduce the results.

### Data

Table 1 describe the input data fields needed by the model. The CoMo model uses two types of data, level measures and parameters; level measures include age specific births, deaths, age specific COVID-19 related hospitalization and deaths and finally daily cases and deaths. Level measures may also include total population, total number of hospital beds and ICU beds. The model uses parameters for Bangladesh specific social contact rates by age, virus, parameters related hospitalization and time varying levels of NPI parameters.

Bangladesh specific age and location specific contact rates developed by Prema et al. [14] were used as inputs.

Virus and hospitalization parameters are taken from published studies. Supplementary S2 Table in S1 File includes sources (references, calibrated or assumed) of all parameters used in this study.

There are 12 NPI parameters that can be used by analyst as intervention measures and for each choice, its level of coverage and date range of its duration will affect levels of projected cases and deaths.

**Table 1. Population, source UN, census 2011; Bangladesh government, SVRS 2018.**

| Source: | UN-Census 2011[a] | | SVRS2018 [b] | |
|---|---|---|---|---|
| Age Group | N | % | N | % |
| 0–4 | 15,061,970 | 10% | 17,049,026 | 8% |
| 5–9 | 18,173,229 | 13% | 20,570,739 | 10% |
| 10–14 | 16,646,615 | 12% | 18,842,726 | 11% |
| 15–19 | 12,861,890 | 9% | 14,558,699 | 10% |
| 20–24 | 13,299,789 | 9% | 15,054,368 | 9% |
| 25–29 | 13,479,508 | 9% | 15,257,797 | 8% |
| 30–34 | 10,499,765 | 7% | 11,884,950 | 8% |
| 35–39 | 9,556,428 | 7% | 10,817,163 | 7% |
| 40–44 | 8,261,662 | 6% | 9,351,585 | 7% |
| 45–49 | 6,380,073 | 4% | 7,221,766 | 5% |
| 50–54 | 5,552,271 | 4% | 6,284,756 | 5% |
| 55–59 | 3,500,997 | 2% | 3,962,867 | 4% |
| 60–64 | 3,934,014 | 3% | 4,453,010 | 3% |
| 65–69 | 2,113,490 | 1% | 2,392,313 | 2% |
| 70–74 | 2,231,712 | 2% | 2,526,131 | 1% |
| 75–79 | 874,727 | 1% | 990,126 | 1% |
| 80–84 | 880,079 | 1% | 996,184 | 1% |
| 85–89 | 262,611 | 0% | 297,256 | 0% |
| 90–94 | 250,189 | 0% | 283,195 | 0% |
| 95 + | 222,678 | 0% | 252,055 | 0% |
| | 144,043,697 | 100% | 163,046,713 | 100% |
| Age Group | <15 | 35% | 35% | 29% |
| Age Group | 65+ | 5% | 5% | 5% |

a: http://data.un.org/Data.aspx?d=POP&f=tableCode%3A22

b:https://bbs.portal.gov.bd/sites/default/files/files/bbs.portal.gov.bd/page/6a40a397_6ef7_48a3_80b3_78b8d1223e3f/SVRS_Report_2018_29-05-2019%28Final%29.pdf

For Bangladesh, age-specific population, births and deaths data as well as family size were estimated using data from Survey of Vital Record (SVR), 2018. The details are in Tables 1–3. For hospitalization, it was assumed that there were 1,000 ICU beds in the country and number of total hospital beds was assumed to be 12,000 [15]. Parameters for virus, other hospital transition probabilities and interventions are changed during the visual calibration process. Levels and parameters assumed in the model are described in Table 4.

The report notes that crude death rate is 5 per 1,000; at a population of 163,046,713, this translates to total annual deaths of about 800,000 but using the age specific annual death rates, total deaths is about 900,000 or 5.8 per 1,000.

## Results

In the context of Bangladesh, the first COVID-19 case was identified on March 8 and by March 23, 2020, the total cases reached 33. It is at that point that the government of Bangladesh announced shutdown of public, private offices, schools and colleges from March 26, 2020 for 10 days [16] (note:the government declared a general holiday/ leave instead of calling it 'lockdown' or 'shutdown'. However the objectives were the same—to tame the spread of the virus.). Subsequently, all public transport systems were shut down [17]. Police, the Army and other security forces were deployed to enforce social distancing. Announcement of the initial

**Table 2. Age Specific Birth Rate (ASBR).**

| | | 2018 | 2018 |
|---|---|---|---|
| | | Est. female | No |
| | | Pop | of Births |
| Age Group | ASFR[a] | = Pop*0.5 | = ASFR*Est. Female Pop |
| 15–19 | 0.07 | 7,279,350 | 538,672 |
| 20–24 | 0.13 | 7,527,184 | 993,588 |
| 25–29 | 0.11 | 7,628,898 | 808,663 |
| 30–34 | 0.06 | 5,942,475 | 356,549 |
| 35–39 | 0.03 | 5,408,582 | 140,623 |
| 40–44 | 0.01 | 4,675,792 | 32,731 |
| 45–49 | 0.00 | 3,610,883 | 10,833 |
| total | 0.41 | 42,073,165 | 2,881,658 |
| TFR | | | 2.04 |

ASFR: Age Specific Fertility Rate

lockdown, in the form of a general holiday, for 10 days was followed by multiple extensions. Since the announcement on March 24, 2020, in several occasions, there have been relaxation of lockdown including allowing garments workers to join work [18], allowing stores to open

**Table 3. Age Specific Death Rate (ASDR).**

| | ASDR[a] | | Total | Total |
|---|---|---|---|---|
| | # per 1,000 | % | Pop | # of Deaths |
| | A | B = A/1000 | C | D = B*C |
| 1 | 27.1 | 2.71% | 17,049,026 | 462,029 |
| 1–4 | 2 | 0.20% | 20,570,739 | 41,141 |
| 5–9 | 0.6 | 0.06% | 18,842,726 | 11,306 |
| 10–14 | 0.5 | 0.05% | 14,558,699 | 7,279 |
| 15–19 | 1.2 | 0.12% | 15,054,368 | 18,065 |
| 20–24 | 0.9 | 0.09% | 15,257,797 | 13,732 |
| 25–29 | 0.9 | 0.09% | 11,884,950 | 10,696 |
| 30–34 | 1.1 | 0.11% | 10,817,163 | 11,899 |
| 35–39 | 1.5 | 0.15% | 9,351,585 | 14,027 |
| 40–44 | 2.2 | 0.22% | 7,221,766 | 15,888 |
| 45–49 | 4.5 | 0.45% | 6,284,756 | 28,281 |
| 50–54 | 6.8 | 0.68% | 3,962,867 | 26,947 |
| 55–59 | 9.4 | 0.94% | 4,453,010 | 41,858 |
| 60–64 | 14.1 | 1.41% | 2,392,313 | 33,732 |
| 65–69 | 23.3 | 2.33% | 2,526,131 | 58,859 |
| 70–74 | 27.5 | 2.75% | 990,126 | 27,228 |
| 75–79 | 81.9 | 8.19% | 996,184 | 81,587 |
| 80+ | 109.3 | 10.93% | 297,256 | 32,490 |
| | | | | |
| Total | 5.8 | | 162,511,463 | 937,047 |
| Total at 5 per 1000 | 5.0 | 0.50% | 163,046,713 | 815,234 |

ASDR: Age Specific Death Rate

**Table 4. Key parameter table\*.**

| Sheet-Intervention | Parameter | Value |
|---|---|---|
| Parameters | (changed in v17) Number of exposed people at start date | 10 |
| Parameters | (v16.2) Proportion of population with partial immunity at the start date | 32 |
| Parameters | Probability of infection given contact (0 to 0.2) | 0.041 |
| Parameters | Percentage of all asymptomatic infections that are reported | 1 |
| Parameters | Percentage of all symptomatic infections that are reported | 1 |
| Country Area Param | Social Contacts Data | Bangladesh |
| Country Area Param | Mean Household size | 4.1 |
| Country Area Param | Mean number of infectious migrants per day | 0 |
| Virus Param | Relative infectiousness of incubation phase | 10 |
| Virus Param | Average incubation period (1 to 7 days) | 3 |
| Virus Param | Average duration of symptomatic infection period (1 to 7 days) | 4 |
| Virus Param | Probability upon infection of developing clinical symptoms | 25 |
| Virus Param | Probability upon hospitalisation of requiring ICU admission | 10 |
| Virus Param | Probability upon admission to the ICU of requiring a ventilator | 50 |
| Virus Param | Proportion of hospitalised patients needing O2 | 50 |
| Hospitalisation Param | Maximum number of hospital surge beds | 130000 |
| Hospitalisation Param | Maximum number of ICU beds without ventilators | 2000 |
| Hospitalisation Param | Maximum number of ICU beds with ventilators | 2000 |
| Hospitalisation Param | Relative percentage of regular daily contacts when hospitalised: | 15 |
| Hospitalisation Param | Scaling factor for infection hospitalisation rate: (0.1 to 5) | 1.1 |
| Hospitalisation Param | Probability of dying when hospitalised (not req O2): | 15 |
| Hospitalisation Param | Probability of dying when hospitalised if req O2: | 20 |
| Self-isolation if Symptomatic | Adherence: | 50 |
| (\*Self-isolation) Screening | Overdispersion: (1, 2, 3, 4 or 5) | 4 |
| (\*Self-isolation) Screening | Test Sensitivity: | 80 |
| (\*Self-isolation) Household Isolation | Days in isolation for average person: | 14 |
| (\*Self-isolation) Household Isolation | Days to implement maximum quarantine coverage: (1 to 5) | 2 |
| (\*Self-isolation) Household Isolation | Decrease in the number of other contacts when quarantined: | 20 |
| (\*Self-isolation) Household Isolation | Increase in the number of contacts at home when quarantined: | 100 |
| Social Distancing | Adherence: | 100 |
| Handwashing | Efficacy: (0–25%) | 20 |
| Mask Wearing | Efficacy: (0–35%) | 15 |
| Working at Home | Efficacy: | 85 |
| Working at Home | Home contacts inflation due to working from home: | 10 |
| School Closures | Home contacts inflation due to school closure: | 20 |
| Shielding the Elderly | Efficacy: | 95 |
| Shielding the Elderly | Minimum age for elderly shielding: (0 to 100) | 70 |
| Vaccination | Time to reach target coverage (1 to 52) | 4 |
| Vaccination | (v16.2) Duration of efficacious period | 100 |
| Vaccination | (v16.2) Duration of efficacious period if previously infected | 100 |
| Vaccination | Efficacy | 100 |
| Vaccination | (v16.2) Efficacy if previously infected | 100 |
| Mass Testing | Sensitivity | 80 |
| Mass Testing | Isolation days | 14 |

and relaxation of travel during Eid religious holiday [19] Finally, the Government announced that all general holidays will end on May 30 and work and travel can resume on May 31; however, there would be a strict enforcement of NPI such as social distancing, and vulnerable population such as the elderly, pregnant women and those with multiple chronic diseases are discouraged to re-join their workplace. Schools and colleges remained closed.

Assumption of interventions in the baseline calibration shows that since the peak of cases in May to July, 2020, the NPI levels have been declining, with most dramatic decline happening in middle of February, 2021.

Notable among the trend are declines in mask-wearing and hand washing where for instance, handwashing dropped from 88% to 0%. Although vaccination was started on February 7,2021 for 40 year old and older and for priority occupations, it is assumed that in the baseline scenario the vaccination will be offered to ages 20+. Based on the recent evidence on vaccination uptake, it is estimated that 3% of the 20+ population will be vaccinated by 2021. In the baseline scenario, schools are assumed to be closed until May 22, 2021 and in hypothetical scenario, schools are assumed to open on May 23, 2021. Both baseline and hypothetical parameters are applied in model for cases up until July 23, 2021.

Based on these events and the corresponding cases and deaths and observations from google mobility data, the NPI parameters were set and adjusted so the model projections closely aligned with actual levels, Eight NPI were selected and these include school closures, hand washing, working at home, self-isolation if symptomatic, social distancing, mass testing, mask wearing and vaccinations.

To account for under reporting, the model allows selection of levels of cases that are reported. For this purpose, model assumed that 1% of cases are reported.

Other important parameters that were changed in the model include percentage of population with partial immunity at start of pandemic was 32%, percent of infection given contact at 4.1%. Virus related parameters that were changed from default levels include duration of incubation period to 3–4 days and key hospital related parameters for infection to hospitalization scaling factor to 1.1.

Using these levels of NPI and assumed virus, hospital parameters, the model predictions were very closely aligned with actual levels.

In the baseline scenario, NPI are assumed to end by May 26, 2021. Vaccination against COVID-19 is expected to reach about 4% of the age-18+ population by December, 2021 (Table 5).

The result of the baseline assumptions indicates that by December, 2021, (Fig 2) there may be another surge of cases in July which will subsequently subside. By end of December, 2021, about 47% of the population will be infected and total deaths will reach about 20,000. The rate of infection reflects reported and un-reported cases. These findings are in-line with findings from other jurisdictions (Table 7).

To test the impact of NPI compliance with and without school openings we construct several hypothetical scenarios and compare it to the baseline scenarios.

Under Hypothetical scenario 1 (see Fig 3), schools are assumed to be closed for one additional months from June 13, 2021 to July 12, 2021 and other NPI parameters are assumed to return to levels observed during the pre-May 26 levels (Table 6). With one additional month of school closure, projected levels of infection rate is expected to decline from 47% to 38% and projected deaths are expected to decline by about 5,000 to 15,000(Table 7).

Under Hypothetical scenario 2 (see Fig 4), schools are assumed to open on June 13; however, the NPI measures such as mask wearing and hand washing are projected to increase from 5% to 10% to abut July 12, 2021. With these higher NPI, infection rates is also expected to decline to 38.2% and deaths are also expected to decline to 15,000.

**Table 5. NPI parameters assumed in baseline scenario (Actual as of May 26, 2021).**

| Intervention | Date Start | Date End | Value | Unit | Age Groups |
|---|---|---|---|---|---|
| School Closures | 3/23/2020 | 6/12/2021 | 100 | % | 5–20 |
| Handwashing | 3/23/2020 | 1/15/2021 | 88 | % | |
| Working at Home | 3/23/2020 | 5/30/2020 | 69 | % | |
| Working at Home | 5/31/2020 | 4/26/2021 | 10 | % | |
| Social Distancing | 3/23/2020 | 6/15/2020 | 32 | % | |
| Social Distancing | 6/16/2020 | 6/30/2020 | 72 | % | |
| Social Distancing | 7/1/2020 | 7/30/2020 | 50 | % | |
| Social Distancing | 7/31/2020 | 10/10/2020 | 52 | % | |
| Social Distancing | 10/11/2020 | 12/15/2020 | 24 | % | |
| Social Distancing | 12/16/2020 | 2/22/2021 | 53 | % | |
| Mask Wearing | 3/23/2020 | 2/20/2021 | 24 | % | |
| Mass Testing | 3/23/2020 | 6/30/2020 | 15 | thousand tests | 18+ |
| Mass Testing | 7/1/2020 | 4/26/2021 | 12 | thousand tests | 18+ |
| Self-isolation if Symptomatic | 3/23/2020 | 2/20/2021 | 18 | % | |
| Self-isolation if Symptomatic | 2/21/2021 | 4/26/2021 | 0 | % | |
| Social Distancing | 2/23/2021 | 4/10/2021 | 0 | % | |
| Mask Wearing | 2/21/2021 | 4/10/2021 | 0 | % | |
| Handwashing | 2/21/2021 | 4/10/2021 | 0 | % | |
| Vaccination | 2/7/2021 | 5/31/2021 | 1 | | 40+ |
| Mask Wearing | 4/11/2021 | 4/20/2021 | 54 | % | |
| Social Distancing | 4/11/2021 | 4/20/2021 | 54 | % | |
| Mask Wearing | 4/21/2021 | 5/10/2021 | 75 | % | |
| Social Distancing | 4/21/2021 | 5/10/2021 | 75 | % | |
| Mask Wearing | 5/11/2021 | 5/26/2021 | 5 | % | |
| Social Distancing | 5/11/2021 | 5/26/2021 | 5 | % | |

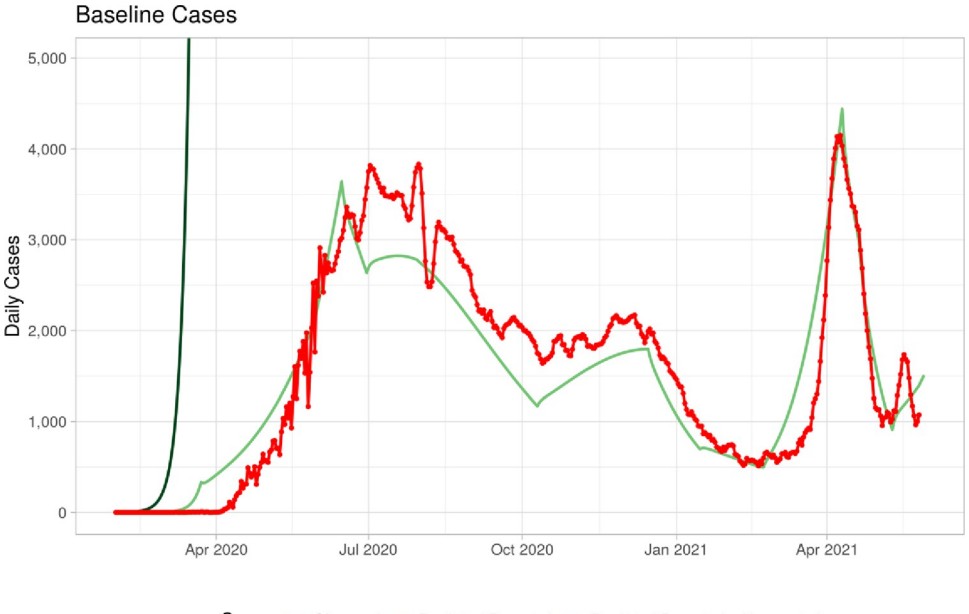

**Fig 2. Baseline calibration.**

**Table 6. NPI Parameters for hypothetical scenarios with and without school opening.**

| | | | Hypothetical 1 | Hypothetical 2 |
|---|---|---|---|---|
| Mask Wearing | 5/27/2021 | 7/12/2021 | 5 | 10 |
| Social Distancing | 5/27/2021 | 7/12/2021 | 5 | 10 |
| Self-isolation if Symptomatic | 5/27/2021 | 7/12/2021 | 5 | 10 |
| Handwashing | 5/27/2021 | 7/12/2021 | 5 | 44 |
| Vaccination | 6/1/2021 | 12/31/2021 | 4 | 4 |
| School Closures | 6/13/2021 | 7/12/2021 | Closed | Open |

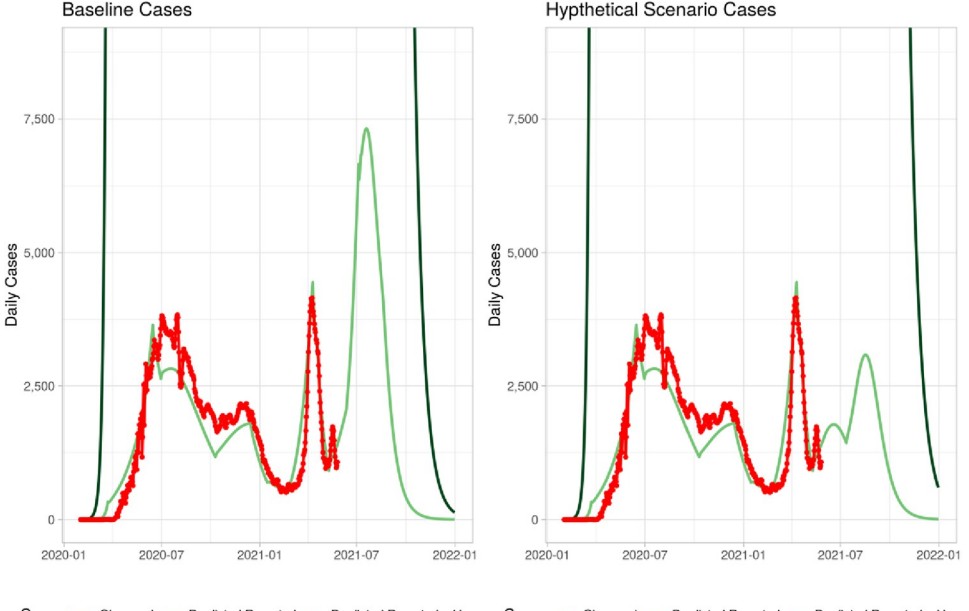

**Fig 3. Hypothetical 1: School closed.**

**Table 7. Hypothetical scenarios.**

| Description | % Population Infected | Deaths |
|---|---|---|
| Baseline: Current NPI | 47.00 | 19,976 |
| Hypothetical 1: School Closed for one additional month, same NPI as Baseline | 38.00 | 15,048 |
| Hypothetical 2: Higher NPI for one additional month and schools open | 38.20 | 15,059 |

Thus, the exercise reveals that potential impact of school opening can be mitigated by increased NPI.

These results support previously found estimates that while school openings can contribute to increased spread of infection; these effects can be mitigated by other NPI.

Several limitations of this study should be noted. Firstly, our model is data dependent. In Bangladesh, infectious disease surveillance does not detect all cases of COVID-19; hence our estimation may be biased by underreporting. Therefore, more accurate data should be put in place to address concerns related to COVID-19. Accurate data leads to better estimates, and conclusions based on these data become more robust. Secondly, we could only adjust for a few crucial parameters in the model. Many other parameters, including age-specific hospitalization

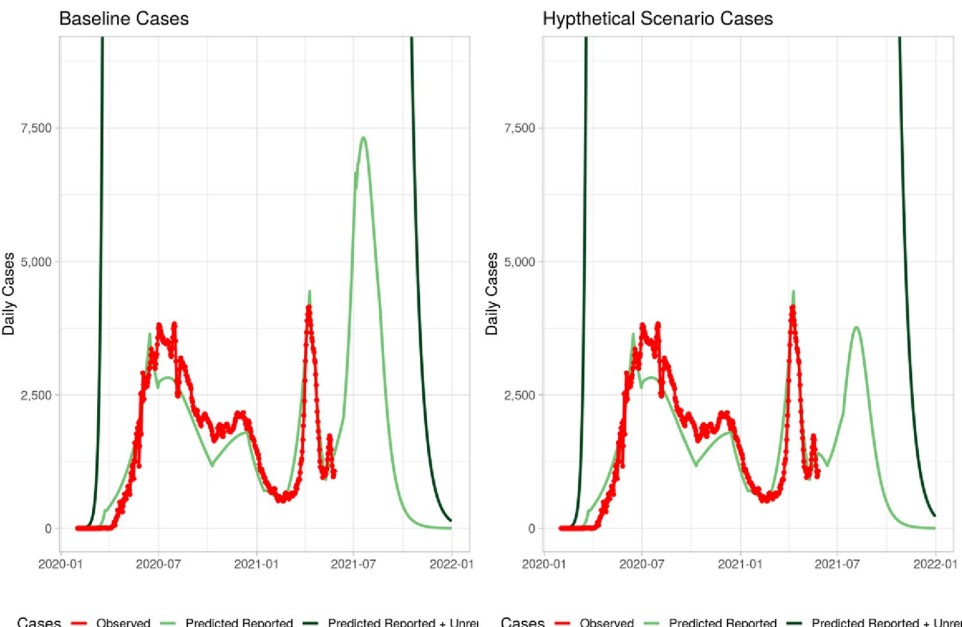

**Fig 4. Hypothetical 2: School open.**

and death rates, were used from published studies from other settings. The analysis could be improved if more input parameters could reflect circumstances in Bangladesh. Further, the model allows for infection rate validation through seroprevalence, a rate not available in Bangladesh. Another approach to improve the current study is to perform division or district-level estimations.

One of the strengths of this study is that this is the first study to explore the effects of NPIs intervention strategies for controlling COVID-19 outbreak in Bangladesh. Second, the outcomes of this study can help with preparedness activities, including formulating a plan and implementing NPIs interventions for the prevention and control of COVID-19.Given these findings the following recommendations are made. First, through coordinated efforts, plans and procedures that are consistent with WHO/UNICEF/UNESCO guidelines should be formulated to open schools. Secondly, all teachers, school administrators and staff should be vaccinated. Thirdly, considerations could be given to a) opening schools in sequences starting with students in lower classes, b) opening schools in communities where incidence and prevalence has been low. Fourthly, parents and broader community should be engaged with sustained communication efforts. Finally, plans to recover learning loss be through extending school year should be discussed.

## Conclusions

Compared to many countries with similar socio-demographic features, Bangladesh's COVID-19 infection rate and the death rate have been low. As of the end of May 2021, the case fatality rate (a measure of deaths per identified COVID-19 cases) in the country was about 1.64, lower than the world average of 2.14 [20]. However, school closures for more than a year may have a significant non-reversible negative impact. In this context, this study shows that strategies for opening schools may be undertaken along with other stronger NPIs to contain the potential negative effects. Recommendations are made to facilitate school opening and reverse some of the learning loss. Due to lack of availability all relevant and country specific data, we had to

make several assumptions to reach in this conclusion. Future research can address these limitations to yield more robust and reliable results.

## Supporting information

**S1 File.**
(DOCX)

**S1 Data.**
(XLSX)

## Author Contributions

**Conceptualization:** Shafiun Nahin Shimul, Mofakhar Hussain, Abul Jamil Faisel, Md Abdul Kuddus.

**Data curation:** Shafiun Nahin Shimul.

**Formal analysis:** Syed Abdul Hamid.

**Investigation:** Mofakhar Hussain.

**Methodology:** Md Abdul Kuddus.

**Software:** Mofakhar Hussain.

**Visualization:** Shafiun Nahin Shimul, Syed Abdul Hamid.

**Writing – original draft:** Shafiun Nahin Shimul.

**Writing – review & editing:** Abul Jamil Faisel, Syed Abdul Hamid, Nasrin Sultana.

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
