## [Decision Letter · Decision Letter 0]

4 Dec 2021

PONE-D-21-27150Impact of alternative Non-Pharmaceutical Interventions strategies for controlling COVID-19 outbreak in Bangladesh: a modelling studyPLOS ONE

Dear Dr. Shimul,

Thank you for submitting your manuscript to PLOS ONE. After careful consideration, we feel that it has merit but does not fully meet PLOS ONE’s publication criteria as it currently stands. Therefore, we invite you to submit a revised version of the manuscript that addresses the points raised during the review process.

We look forward to receiving your revised manuscript.

Kind regards,

Kin On Kwok, Ph.D

Academic Editor

PLOS ONE

Journal Requirements:

3. Please ensure that you refer to Figure 3 and 4 in your text as, if accepted, production will need this reference to link the reader to the figure.

4. We note you have included a table to which you do not refer in the text of your manuscript. Please ensure that you refer to Table 2 and 3 in your text; if accepted, production will need this reference to link the reader to the Table.

Additional Editor Comments:

Authors are recommended to give a brief introduction on the como model in the method section and present how they applied their data on the CoMo model. Given the epidemic dynamics is likely to be shaped by contact heterogeneous mixing across different age groups as we observed in other populations (https://pubmed.ncbi.nlm.nih.gov/29367241/), your model may not be sufficient to generate the accurate transmission dynamics of COVID-19. I suggest to include this limitation in your discussion. English editing is required to improve the language of the manuscript before it is accepted for publication.

Reviewers' comments:

Reviewer's Responses to Questions

**Comments to the Author**

1. Is the manuscript technically sound, and do the data support the conclusions?

Reviewer #1: Partly

Reviewer #2: Yes

2. Has the statistical analysis been performed appropriately and rigorously? 

Reviewer #1: No

Reviewer #2: Yes

3. Have the authors made all data underlying the findings in their manuscript fully available?

Reviewer #1: Yes

Reviewer #2: Yes

4. Is the manuscript presented in an intelligible fashion and written in standard English?

Reviewer #1: Yes

Reviewer #2: No

5. Review Comments to the Author

Reviewer #1: Abbreviations of Table 2 and Table tow are same (ASDR). Please correct them.

Please clarify the figure 800 000 million in line 66 of introduction section.

Authors have not shown how they have applied CoMo method for their data. Although the tables include statistics, the authors have not produced a detailed statistical analysis about the data. Also, authors should describe CoMo model under methods section.

Column headings of the tables are not clear.

the authors clam that the death rate is low in Bangladesh compared to the other countries (having similar socioeconomic conditions). However, they have not provided any comparison (in tabular format). Therefore, it is hard to accept the authors claim.

Language errors must be improved.

Reviewer #2: The paper used the web-based Como model built by CoMo consortium, so the method is supposed to be sound.

Data source and parameter inputs are provided clearly.

However, the manuscript is not really well-written, with sudden line breaks, making it sometimes difficult to follow.

6. PLOS authors have the option to publish the peer review history of their article (what does this mean?). If published, this will include your full peer review and any attached files.

Reviewer #1: No

Reviewer #2: No

---

## [Author Response · Author response to Decision Letter 0]

7 Jul 2022

Please see the attached files for detailed response

---

## [Decision Letter · Decision Letter 1]

8 Nov 2022

PONE-D-21-27150R1Impact of alternative Non-Pharmaceutical Interventions strategies for controlling COVID-19 outbreak in Bangladesh: a modelling studyPLOS ONE

Dear Dr. Shimul,

Thank you for submitting your manuscript to PLOS ONE. After careful consideration, we feel that it has merit but does not fully meet PLOS ONE’s publication criteria as it currently stands. Therefore, we invite you to submit a revised version of the manuscript that addresses the points raised during the review process.

The manuscript lacks a certain level of technical detail, in particular, a careful mathematical representation of the model used to carry out the analysis. Furthermore, the manuscript needs to be thoroughly proofread for language.

We look forward to receiving your revised manuscript.

Kind regards,

Leonid Chindelevitch, Ph.D.

Academic Editor

PLOS ONE

Journal Requirements:

Reviewers' comments:

Reviewer's Responses to Questions

**Comments to the Author**

1. If the authors have adequately addressed your comments raised in a previous round of review and you feel that this manuscript is now acceptable for publication, you may indicate that here to bypass the “Comments to the Author” section, enter your conflict of interest statement in the “Confidential to Editor” section, and submit your "Accept" recommendation.

Reviewer #1: All comments have been addressed

2. Is the manuscript technically sound, and do the data support the conclusions?

Reviewer #1: Partly

3. Has the statistical analysis been performed appropriately and rigorously? 

Reviewer #1: Yes

4. Have the authors made all data underlying the findings in their manuscript fully available?

Reviewer #1: Yes

5. Is the manuscript presented in an intelligible fashion and written in standard English?

Reviewer #1: No

6. Review Comments to the Author

Reviewer #1: Still, there are plenty of language-related errors. I recommend authors use proofreading tools such as Grammarly to check the language errors.

In the Data section, line 182 is an incomplete sentence. Several such mistakes can be found in the document.

Although the authors have explained the CoMo model, it is not sufficient. Please include a technical explanation of the model (mathematical explanation).

7. PLOS authors have the option to publish the peer review history of their article (what does this mean?). If published, this will include your full peer review and any attached files.

Reviewer #1: No

---

## [Author Response · Author response to Decision Letter 1]

3 Jan 2023

Dear Reviewer, Thank you so much for your insightful feedback. We tried to address all of you concerns. Please find the attached document for details.

---

## [Editor Report · Decision Letter 2]

3 Feb 2023

PONE-D-21-27150R2Impact of alternative Non-Pharmaceutical Interventions strategies for controlling COVID-19 outbreak in Bangladesh: a modeling studyPLOS ONE

Dear Dr. Shimul,

Thank you for submitting your manuscript to PLOS ONE. After careful consideration, we feel that it has merit but does not fully meet PLOS ONE’s publication criteria as it currently stands. Therefore, we invite you to submit a revised version of the manuscript that addresses the points raised during the review process.

We look forward to receiving your revised manuscript.

Kind regards,

Leonid Chindelevitch, Ph.D.

Academic Editor

PLOS ONE

Journal Requirements:

Additional Editor Comments (if provided):

Dear authors,

Despite the second round of revisions there are still significant issues remaining with the manuscript, which will make it impossible for me to accept it in its current form.

1) The mathematical equations you have provided are all in the form dX/dt = Function(variables); this is not acceptable for reproducibility reasons. The only way this paper can be accepted is if you explicitly provide all of the functions. Please see some modelling papers for examples, such as https://www.nature.com/articles/s41586-020-2405-7, where all the code is provided, or at the very least https://pubmed.ncbi.nlm.nih.gov/33323424/, where all the equations are listed explicitly. Without either of these the paper will be rejected.

2) The proofreading is not complete; there are still a number of issues with spacing, leftover characters (tJune at the start of the document for example), and formatting problems. Please make sure that the paper looks presentable for publication and is free of errors to the best of your ability.

Unfortunately, I will have to reject it if your next revision does not comply with these requirements.

Thanks,

The editor.
---

## [Author Response · Author response to Decision Letter 2]

13 May 2023

We, hereby, are resubmitting our research article for your kind perusal. In this study we attempted to understand the impact of school opening on COVID-19 outbreak for Bangladesh. Editor has expressed concern on the replicability and suggested either writing detailed model structure or by providing steps to be taken to replicate. 

We have since discussed the matter of ‘detailed model structure’ with the original authors of the model used in our study. We were advised to refrain from discussing details of the model as we are not the original authors. 

Following the advice we have revised our manuscript and removed detail of the model. However, in a Supplementary, we have provided reference to the GitHub site where details of the model can be found. We have also provided steps to access the model and input data sets used in our study. The model along with the input data sets (templates) can be used to re-create the results reported in our study. It is hoped that this material will address the concern raised regarding model details in our study.

We look forward to your response.

---

## [Editor Report · Decision Letter 3]

18 Jun 2023

PONE-D-21-27150R3Impact of alternative Non-Pharmaceutical Interventions strategies for controlling COVID-19 outbreak in Bangladesh: a modeling studyPLOS ONE

Dear Dr. Shimul,

Thank you for submitting your manuscript to PLOS ONE. After careful consideration, we feel that it has merit but does not fully meet PLOS ONE’s publication criteria as it currently stands. Therefore, we invite you to submit a revised version of the manuscript that addresses the points raised during the review process.

We look forward to receiving your revised manuscript.

Kind regards,

Leonid Chindelevitch, Ph.D.

Academic Editor

PLOS ONE

Journal Requirements:

Additional Editor Comments:

Dear authors,

Thank you for addressing the comments. As the model-producing tool is not one that you created yourselves, but rather the Oxford team, please make sure that you:

a) get their written agreement to share the model details (unless it was publicly available) and upload that as a supplementary file,

b) reference all the relevant literature sources used for calibrating the model's parameters,

and c) clarify the relative contributions of yours relative to the Oxford team in the manuscript (this was not very clear until now, even after I read the paper twice).

Once these changes have been implemented the paper should be ready.

Your academic editor
---

## [Author Response · Author response to Decision Letter 3]

2 Oct 2023

Thank you so much for your insightful feedback.

---

## [Editor Report · Decision Letter 4]

23 Oct 2023

Impact of alternative Non-Pharmaceutical Interventions strategies for controlling COVID-19 outbreak in Bangladesh: a modeling study

PONE-D-21-27150R4

Dear Dr. Shimul,

We’re pleased to inform you that your manuscript has been judged scientifically suitable for publication and will be formally accepted for publication once it meets all outstanding technical requirements.

Kind regards,

Leonid Chindelevitch, Ph.D.

Academic Editor

PLOS ONE

Additional Editor Comments (optional):

Thanks for addressing my remaining requests. I only have one minor typo correction - instead of "Someone can reproduce the results" it should be "Anyone can reproduce the results".
---

## [Editor Report · Acceptance letter]

5 Dec 2023

PONE-D-21-27150R4 

Impact of alternative Non-Pharmaceutical Interventions strategies for controlling COVID-19 outbreak in Bangladesh: a modeling study 

Dear Dr. Shimul:

I'm pleased to inform you that your manuscript has been deemed suitable for publication in PLOS ONE. Congratulations! Your manuscript is now with our production department. 

Kind regards, 

on behalf of

Dr. Leonid Chindelevitch 

Academic Editor

PLOS ONE